# The lived experiences of UK physiotherapists involved in Cauda Equina Syndrome litigation. A qualitative study

Gillian Yeowell[1]*, Rachel Leech[1], Susan Greenhalgh[1,2], Emma Willis[1], James Selfe[1]

1 Department of Health Professions, Faculty Health and Education, Manchester Metropolitan University, Manchester, United Kingdom, 2 Bolton NHS Foundation Trust, Orthopaedic Interface Service, Bolton One, Bolton, Manchester, United Kingdom

☯ These authors contributed equally to this work.

* g.yeowell@mmu.ac.uk

**Data Availability Statement:** Data supporting this publication are openly available from Manchester Metropolitan University's research repository: DOI 10.23634/MMU.00632413.

## Abstract

### Background

Cauda Equina Syndrome is a serious spinal pathology, which can have life changing physical and psychological consequences and is highly litigious. Litigation can have negative personal and professional effects on the healthcare professionals cited in a clinical negligence claim. There is an absence of research looking at the experience of the physiotherapist and as such, it is unknown the impact litigation is having on them. This study explored the lived experiences of UK physiotherapists in relation to Cauda Equina Syndrome litigation.

### Methods

A qualitative design, informed by Gadamerian hermeneutic phenomenology, using semi-structured interviews was used to explore participants' lived experiences of litigation. Interviews were audio-recorded and transcribed verbatim. Findings were analysed using an inductive thematic analysis framework. Nvivo software was used to facilitate analysis. The study is reported in accordance with the consolidated criteria for reporting qualitative (COREQ) research.

### Results

40 interviews took place online or over the phone, with physiotherapists and stakeholders. Four themes were found; 'litigation effects', 'it feels personal', 'learning from litigation' and 'support and training'.

### Conclusion

This is the first study to investigate the lived experiences of litigation in UK physiotherapists. Involvement in clinical negligence affected physiotherapists' physical and mental wellbeing and impacted their clinical practice. Most physiotherapists felt litigation was a personal attack on them and their ability to do their job. Physiotherapists highlighted perceptions of a

**Funding:** GY, SG, JS, EW received a grant to support this research from the Chartered Society of Physiotherapy Charitable Trust (Grant number PRF/19/A18). https://www.csp.org.uk/about-csp/how-we-work/charitable-trust The funder played no role in the study design, data collection and analysis, decision to publish, or preparation of the manuscript.

**Competing interests:** The authors have declared that no competing interests exist.

'blame culture' and perceived stigma associated with the claim, which often led to a lack of sharing and learning from litigation. Physiotherapists emphasised the need for emotional support for those going through a legal claim and that training was needed to understand the process of litigation and range of potential outcomes.

## Introduction

Cauda Equina Syndrome (CES) is a rare, yet serious spinal pathology potentially requiring emergency spinal surgery. The incidence of CES ranges from 0.3 to 7.0 per 100,000 population per year [1]. Poor outcomes of CES can occur if the condition is not managed in a timely manner, which can have life changing physical and psychological consequences [2]. Due to this, CES is highly litigious, costing the National Health Service (NHS) in the United Kingdom (UK) in excess of 186 million over a 10 year period (2008–2018) [3].

Litigation in healthcare varies worldwide due to different legal and healthcare systems. Like the UK, the United States of America has an adversarial system of medical malpractice claims, although American states have different regulations related to medical negligence [4]. New Zealand, however, changed their legal system in 2005 to a less adversarial approach to clinical negligence, moving from a punitive system to one that encouraged learning [5]. In the UK, physiotherapists are increasingly being cited in clinical negligence cases, which may be related to their changing role. The role of the physiotherapist differs around the world, including their degree of autonomy and professional rights. Within a UK context, whilst physiotherapists have possessed professional autonomy for many years the role is rapidly changing with increasing numbers of advanced roles, including advanced practice physiotherapy (APP) and first contact practitioner (FCP) roles, in settings such as emergency departments and Primary Care [6]. Therefore, more physiotherapists are likely to become a first point of contact for patients who have not been screened by a medic for a serious pathology such as CES, and consequently are becoming increasingly involved in litigation claims [7, 8]. Previous research found a total of 2496 CES claims in the UK between 2012–2020 [7, 8]. Of these, 51 were attributed to physiotherapists, however this is likely to be an underestimation due to deficiencies in current reporting methods [8].

Whilst the impact of a patient safety incident on the patient and family is the key concern, for the healthcare professional (HCP) involved in a clinical negligence claim there is growing recognition of the impact it can have on them [8]. The HCP can experience a significant impact, both personally and professionally and consequently have been described as 'second victims' [9]. It has been found in other HCP that being involved in a clinical negligence claim can lead to loss of confidence, self-doubt and absence from work [9, 10], and for some, it can lead to them considering changing jobs to work in areas of clinical practice not considered as high-risk of litigation or to leave their profession all together [10].

To protect themselves from liability, HCPs may adopt defensive practice [11]. Defensive practice refers to the over-cautious management of patients, leading to excessive clinical activity including over-investigation and additional interventions and deviating away from what may be considered best practice in order to protect themselves from liability rather than advancing the care of patients [11, 12]. Defensive practice has been observed in medics and other HCP [10, 13]. It is argued that this is not advantageous to the patient or clinician, as it not only impacts costs in healthcare but the quality of the healthcare system. Furthermore, patients could be exposed to unnecessary and often invasive procedures [13]. In order to reduce harm and prevent claims from happening in the future healthcare organisations,

including the NHS, need to learn from things that go wrong [14]. When investigating the effects of litigation, it has been suggested that learning from litigation claims can help to improve patient safety [14].

Most research has focussed on the experiences of litigation among medics and midwives. There is an absence of research looking at the experience of the physiotherapist and as such, it is unknown the impact litigation is having on them. The current study aimed to explore the lived experiences of UK physiotherapists in relation to CES and litigation.

## Methods

The study is reported in accordance with the consolidated criteria for reporting qualitative (COREQ) research [15]. Ethical approval was obtained from Manchester Metropolitan University Faculty Ethics Committee, UK (Ref: 18122). Informed consent was obtained verbally from all participants prior to participation. Participant consent was digitally audio-recorded by the interviewers.

A qualitative design, informed by Gadamerian hermeneutic phenomenology was used to explore participants' experiences of litigation. Semi-structured interviews were undertaken to reveal meaning through a process of understanding and interpretation, thereby addressing the research aim [16].

Participants were purposively recruited through professional networks between January-July 2021. Snowball sampling was used to recruit further participants [17]. Recruitment continued until data saturation was achieved [17]. Participants were eligible if they were a qualified physiotherapist who had been involved in CES litigation in the UK. However, to gain a holistic understanding of the phenomenon of interest, previous research indicated the need to understand the experience of UK physiotherapists at risk of CES litigation i.e., those working in advanced roles [7, 8]. Furthermore, for context and to aid understanding of the issue, we also needed to speak to a range of stakeholders, who included:

i. Other HCPs with experience of litigation

ii. Legal people involved in the litigation process

iii. Representatives of HCP professional bodies

iv. Clinical leads

In-depth one to one semi-structured interviews were undertaken using Microsoft Teams or via telephone with a member of the research team who was experienced in qualitative interviewing (GY, SG, RL), two of whom were physiotherapists, the third was a research assistant. To ensure interviews were conducted consistently between researchers, two interviewers were normally present in each interview, with one conducting the interview and a second interviewer listening and making field notes (on mute, with their camera turned off). In the initial stages of data generation, due to the volume of interviews, additional support was provided by a research associate experienced in qualitative interviewing. Participants were unknown to the interviewers or known in a professional capacity as physiotherapists. Only the research team interviewers had access to information that could identify individual participants during or after data collection. All interviewers listened to the audio-recordings, read the transcripts, and met regularly throughout data generation to reflexively discuss the interview. An interview topic guide (S1 File) was used to guide the interview to provide further consistency and to direct the interview by providing a priori topics to be explored in relation to the aim of the study, whilst allowing sufficient flexibility to explore new and unanticipated issues. The interview guide was developed from a review of the literature [7, 8] and was refined following

piloting and critical discussion with the research team and a patient and public involvement group (PPI). The PPI group included three people living with CES and a physiotherapist who had been involved in a CES litigation case. Interviews lasted between 60 and 90 min and were digitally audio-recorded. Interviews were transcribed verbatim by a professional transcriber to ensure accuracy of the transcription.

Data analysis was undertaken using Braun and Clarke's six phase framework for thematic analysis using an iterative and inductive approach to transform the data [18]. This involved the team (GY, SG, RL) independently listening to the audio-recordings and reading the transcripts. Open coding was used to code the data. This involved reading each transcript line by line to identify salient text related to the research question. Data derived codes were used to summarise the data. Codes were recorded using Nvivo software (version 20.6.1). Patterns across the dataset were then iteratively explored, and theoretically cognate codes were grouped to create sub-themes. Conceptually similar sub-themes were grouped together into emergent themes independently by the research team. The themes were then discussed, critically reviewed, and refined by the research team (GY, SG, RL, EW, JS) to create the final themes. There was concordance in the themes identified by the team and any refinement of themes related to semantics. Preliminary analysis was undertaken after each interview, which iteratively fed into subsequent data generation. Reflexive field notes of the interviewer's role and how this may have impacted on the data generated were made and fed into the analysis of the findings. Member checking was used to validate the findings and ensure the participants' experience of CES litigation were represented and not biased by the researchers' own thoughts and knowledge [15]. Participants confirmed the findings reflected their experiences.

## Results

Forty participants were interviewed. Seventeen participants were physiotherapists who had experience of being involved in a CES litigation case, some of whom were involved in more than one case (Table 1). Eleven participants were physiotherapists at risk of being involved in litigation due to their role involving them being the first point of contact for patients with CES. Twelve participants were stakeholders. These included other HCP with experience of litigation, legal stakeholders who were involved in the litigation process, representatives of HCP professional bodies and clinical leads.

### Themes

Analysis of the data confirmed data saturation had been achieved. Four themes were identified from the data: 'Litigation effects', 'It feels personal', 'Learning from litigation' and 'Support and training' each of which were associated with several sub-themes (Fig 1). Anonymised verbatim quotes have been included to support each theme.

**Theme 1: 'Litigation effects'.** **'Litigation effects'** describes the direct effects of litigation on a physiotherapists' health and wellbeing and encompasses the impact on their clinical practice.

**Litigation effects: Health and wellbeing.** Physiotherapists described the impact on their mental and physical health over the period of their litigation case, which commonly lasted around 2 years. Across the physiotherapists, this included stress, anxiety, insomnia, nausea, high blood pressure, gastric reflux, and a loss of appetite.

> *"I felt sick, I couldn't sleep, . . . I had to go on high blood pressure tablets for some time. I got gastric reflux, which was really bad, it affected my appetite."* (P1, physiotherapist with experience)

**Table 1. Participant demographic data.**

| | Physiotherapist with experience of CES litigation (n = 17) | | Physiotherapist at risk of CES litigation (n = 11) | | Stakeholders (n = 12) |
|---|---|---|---|---|---|
| Number of claims | Claims per participant<br>1 case<br>2 cases<br>3 cases<br>4 cases | Mean n = 1.5 (SD 0.9)<br>n = 12<br>n = 2<br>n = 2<br>n = 1 | NA | | *Other HCPs with experience of litigation*<br>Midwives, medics<br>*Legal*<br>Legal advisors from legal firms; MLACP; expert witness; NHS claims co-ordinators, NHS Resolution<br>*Healthcare professional bodies*<br>National healthcare improvement advisors; CSP representatives; national back pain clinical network representatives, CES national pathway representatives<br>*Clinical leads*<br>NHS physio managers; Clinical and operational leads; Clinical directors non-NHS, Clinical directors AHP NHS |
| Employment category | NHS<br>SE<br>Non-NHS | n = 16<br>n = 5<br>n = 4 | NHS<br>SE<br>Non-NHS | n = 8<br>n = 5<br>n = 0 | |
| Physiotherapy role | Consultant<br>Clinical lead<br>FCP<br>APP<br>AFC Band 7<br>SE /non-NHS | n = 5<br>n = 2<br>n = 1<br>n = 8<br>n = 1<br>n = 2 | Consultant<br>Clinical lead<br>FCP<br>APP<br>SE /non-NHS | n = 5<br>n = 2<br>n = 4<br>n = 2<br>n = 2 | |
| Years qualified | Mean = 24 years (SD 7.83) | | Mean = 25 years (SD 7.69) | | |
| Years in MSK practice | Mean = 20 years (SD 4.96) | | Mean = 23 years (SD 8.22); range | | |
| CES training completed | Extensive<br>MSc Units<br>CPD | n = 5<br>n = 2<br>n = 9 | Extensive<br>MSc Units<br>CPD | n = 2<br>n = 3<br>n = 6 | |
| Litigation training completed | CPD<br>BSc<br>None | n = 6<br>n = 1<br>n = 9 | CPD<br>None | n = 7<br>n = 4 | |

MSK = musculoskeletal, SE = self-employed, FCP = first contact practitioner, APP = advanced practice physiotherapist, CPD = continuing professional development, AFC = agenda for change; CSP = Chartered Society of Physiotherapy, MLACP = Medico Legal Association of Chartered Physiotherapists; AHP = Allied Health Professionals

*"I lost sleep over it. I was just distraught really to be honest. It was really harrowing . . . for two years. . . . Just the anxiety of remembering it, just awful".* (P2, physiotherapist with experience)

This led to some 'taking time off work' due to sickness, 'turning to alcohol' (p1), and changing their role or retiring.

*"Within six months, I'd wanted to go part time, and if they weren't going to give me part time, I don't know what I would've done. There's a possibility that I would have had to quit"* (P33, physiotherapist with experience)

## Litigation effects: Clinical practice implications

Participants told how being involved in litigation had affected their professional confidence.

*"I just didn't know if I was really any good anymore. It had a huge impact on my self-confidence".* (P2, physiotherapist with experience)

**Sub-themes**                                              **Themes**

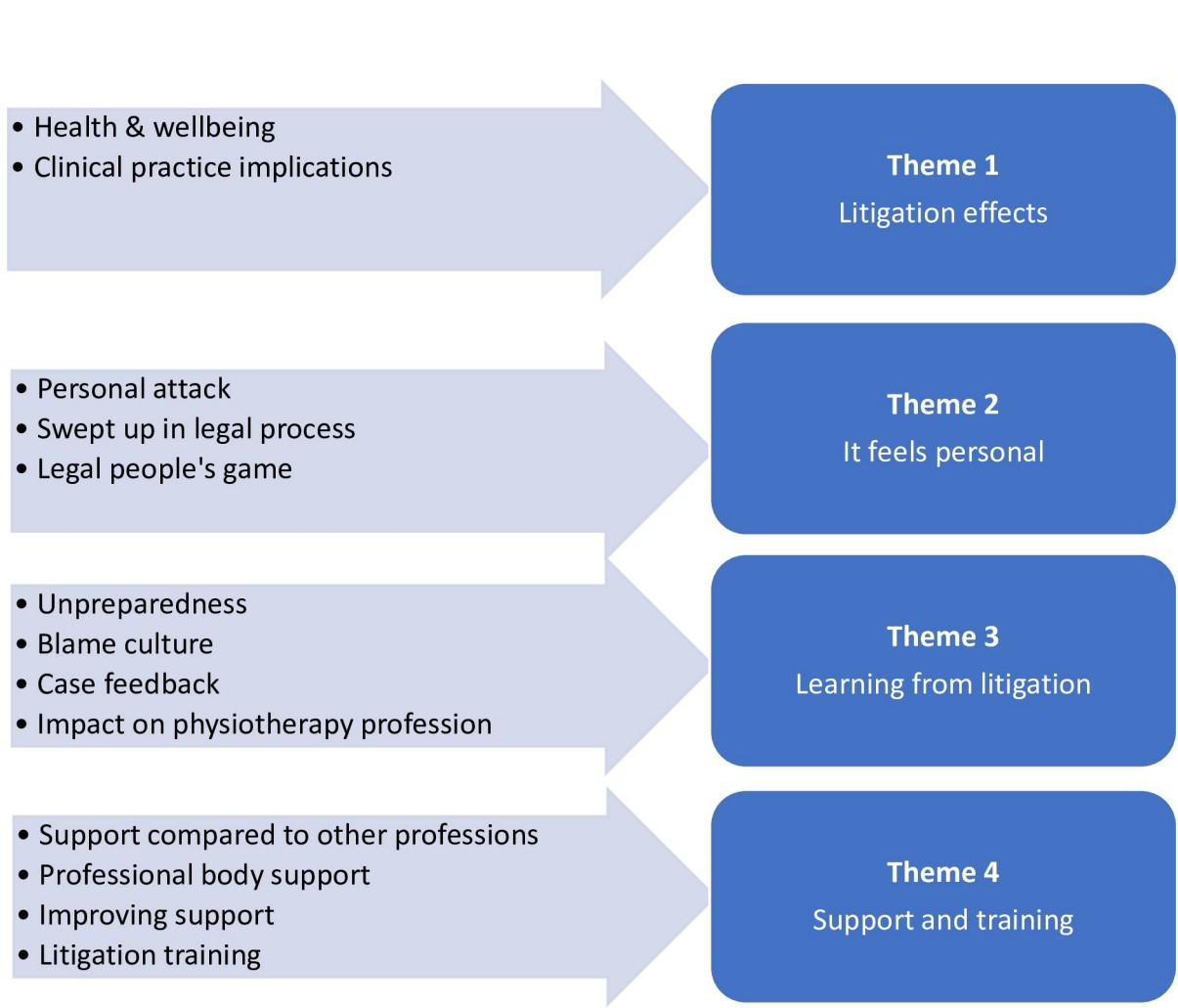

**Fig 1. Themes and sub-themes.**

Those with an experience or an awareness of litigation described the impact of that on the way they managed patients, with many feeling the need to practice in a defensive manner to avoid being cited in a legal claim.

*"It's about, "How do we not get sued?" rather than, "Let's treat the patient using the very best of me and my knowledge and skills, and the very best evidence". . . . We shouldn't really be thinking, "Okay, let's not get sued" first–which is a crying shame."* (P3, physiotherapist at risk)

As a result of litigation, some physiotherapists discussed lowering their thresholds for sending patients for further investigations due to the worry of missing a serious pathology.

*"She said, "Well how has it changed your practice?" I said, "I scan everybody." My threshold to scan was so low because I was so worried about getting this wrong."* (P2, physiotherapist with experience)

Others talked about changes made to their documentation since being involved in a CES claim:

*"I think it has changed my practice. I am a lot more aware of how I'm wording my notes and things like that, and the detail that I am going into with all the notes as well"* (P15, physiotherapist with experience)

Even physiotherapists who did not have their own personal experience of litigation often expressed their awareness of it and described how it impacted their practice.

*"I think I do over-assess, and I over-examine, and I over-document, and that puts on a lot of stress and anxiety [on me]."* (P3, physiotherapist at risk)

### Theme 2: It feels personal

**'It feels personal'** describes how most physiotherapists felt litigation was a personal attack on them and their ability to do their job and described feeling that the process was a personal criticism of their professional ability. Some described a perception of being 'swept up' in the legal process as one of several health professionals involved in the patient journey investigated *en masse* as part of the claim.

### It feels personal: Personal attack

Several physiotherapists with experience of litigation described feeling like litigation was a personal attack on their personal integrity and their ability to do their job:

*"I tried to do everything I could for this patient. I bent over backwards for this patient, and then suddenly I'm faced with this litigation. It feels very, very, very personal. It feels like this is a direct insult on my ability, on my integrity or my ability to do what I'm designed to do in terms of examining patients and dealing with patients. So it feels incredibly personal."* (P2, physiotherapist with experience)

### It feels personal: Swept up in legal process

Physiotherapists can be cited in a complex litigation case regardless of whether they perceive that have been negligent or not.

*"We would obtain the medical records and then I would look at the medical records and I would do a chronology of care. So, we weren't just looking for necessarily where the enquirer thought things had gone wrong, we were looking where we thought things had gone wrong."* (P5, stakeholder legal)

Physiotherapists who came to understand the litigation process due to their previous experience of being involved in a clinical negligence case, realised that litigation was not a personal attack on them. Instead, they realised that when a case is pursued by a claimant, every clinician in the pathway is investigated, making it feel less personal.

*"I became more aware that it's a legal process where the whole pathway is looked at and everybody is swept into it."* (P1, physiotherapist with experience)

### It feels personal: 'Legal people's game'

Several participants described the legal process and how physiotherapists perceived that legal representatives did not understand the complexity of their job:

> *"The lawyers want black and white, and they think it's black and white because they don't understand it [CES]."* (P11, physiotherapist with experience)

> *"I think that it's very unusual that patients present with black-and-white symptoms. Patients–nine times out of ten–will have other co-morbidities or mental health issues, and lots of other things that add to the complexity and that adds to the uncertainty within my daily job."* (P3, physiotherapist at risk)

This perception of the complexity of patient care was supported by other HCP who had experience of litigation:

> *"This is what upsets me about the litigation, the legal teams—they just see it as so black and white. They don't understand. Unless you're that person, in that situation at that moment in time, you just can't understand what's going on in that moment or the emotions, the pressures, the responsibilities and the decision that will have been made at that time. There's never ever going to be any malice or anything like that. It's just so disheartening really."* (P4, stakeholder–HCP with litigation experience)

Physiotherapists also discussed experiences where solicitors described the legal process as sometimes being considered as a 'game' in the context of not taking it personally and reassuring them that if the claim was successful, it would be settled.

> *"[solicitor said] "Don't worry. We'll settle out of court.""* (P1, physiotherapist with experience)

> *"He [solicitor] was going, "You might as well stop crying. This is a game to me, you know." And he was lovely."* (P34, physiotherapist with experience)

### Theme 3: Learning from litigation

'**Learning from litigation**' –In this theme most physiotherapists highlighted a reticence to talk about litigation and to share findings due to perceptions of a 'blame culture' and perceived stigma associated with the claim. Participants perceived this was also impaired by a lack of means by which to share learning more widely. This theme also describes the lack of knowledge around the process and outcome of litigation.

### Learning from litigation: Unpreparedness

Physiotherapists' voiced feelings of their initial reaction to a litigation claim and throughout the course of the case. Physiotherapists unanimously described feeling sheer shock and panic, worrying about the consequences the claim may have for their career and ultimately their ability to provide for their family.

> *"I think because I had not had any experience, or training about it, it's quite a scary situation. You're worrying about, 'Am I going to get struck off? What have I done? What are the*

*implications for it?' So, yes, there is a large fear there really.”* (P15, physiotherapist with experience)

It was highlighted by most physiotherapists that the lack of knowledge about the process of litigation and the possible outcomes exacerbated the stress and anxiety they experienced.

*“It was very stressful because of the wording that was used, that you have been negligent, and those are very strong words. So yes, I mean a whole lot of emotions—the fear, the worry, the doubt, the unknown, I think. A big thing is the unknown, you don't know what I need to do next and what's going to happen, what's likely to happen but yes, it was very, very stressful, a lot of anxiety.”* (P20, physiotherapist with experience)

Physiotherapists also expressed their confusion of where they should go for support with the litigation process and who they were allowed to discuss their case with, which added to their stress.

*“So in that minute of opening the letter when your hands are shaking, what do you do? Can you speak to people about it or is this confidential?* (P1, physiotherapist with experience)

*“That just added to the stress, I think if it had been made clear to me that as an employee of the Trust [hospital], the Trust will cover you. I think if that had been made clear, that would have helped but that was never made clear to me and I felt, probably angry towards the end, that I hadn't had that information because that would have made a huge difference”.* (P20, physiotherapist with experience)

## Learning from litigation: Blame culture

Physiotherapists talked about the stigma surrounding litigation, with many feeling embarrassed, ashamed and even blamed in their workplace for being cited in a claim.

*“It was embarrassing and painful and all those things, really.”* (P11, physiotherapist with experience)

Some explained the importance of having a no-blame culture to facilitate feedback and learning from the claim. Adding that feedback should include both positive and negative experiences to be effective.

*“We have a no-blame culture in work. . . . We look at the whole system. We look at how we can improve things. And we want staff to be able to feel that we can share patients that have gone well and not gone well. And not feel like people are going to think that they're a rubbish physio because, you know, it's not the case.”* (P38, physiotherapist with experience)

## Learning from litigation: Case feedback

Physiotherapists talked about what they learnt through being involved in litigation and how they can use their experiences to make positive changes going forward.

*“A positive impact was that I fed back to the department about the case and what we had learnt from the case, and how we may be able to change sort of future practice, and I think we got a lot tighter with the documentation as a result.”* (P15, physiotherapist with experience)

Some legal stakeholders reflected on how feedback from litigation cases can help to make improvements in care.

*"[We undertake] what's known as a root cause analysis. So once the claim has finished, the outcomes are sent back to the service, so there'll be learning from it. So, the managers can have a look and go, "Oh, there is a gap. We need to do something about that," so that they can stop it from happening again."* (P21, stakeholder legal)

However, many organisations were reticent to talk about litigation or share experiences because of the stigma attached to litigation.

*"There are [organisations] just very much fearful that they don't want to share things because it looks bad on them"* (P28, stakeholder legal)

## Learning from litigation: Impact on physiotherapy profession

Many physiotherapists and other stakeholders perceived CES litigation to be increasing within healthcare, with some participants describing it as the 'new whiplash', which historically has been highly litigious.

*"It's like the new whiplash."* (P40, physiotherapist with experience)

Physiotherapists added that they think that litigation will continue to increase due to advanced roles giving physiotherapists more autonomy and responsibility in the context of patients with complex, uncertain clinical presentations.

*I think it's probably going to get more and more common given that physios are seeing more of this type of patient because the doctors are seeing less of it."* (P40, physiotherapist with experience)

The increasing number of high value litigation claims such as CES claims was reported to have affected insurance premiums. A legal stakeholder suggested that CES litigation may pose a risk against physiotherapists' public liability insurance (PLI), as a single claim in the future could exceed their current cover. They added that physiotherapists could see increases on their insurance premiums as a result.

*"CES is a risk against the PLI because a single claim could, in the future, exceed the current cover of 7.5 million. And that could have a negative impact on [the insurance] premium".* (P39, stakeholder professional body)

## Theme 4: Support and training

'**Support and training**' –In this theme physiotherapists described the support needed for those going through litigation, including emotional support, and having a safe place to talk about any worries relating to the claim. It also explores training that may be needed in relation to litigation during the physiotherapists' career.

## Support and training: Compared to other health professions

Many physiotherapists reflected on the experiences of their colleagues of other professions such as GPs and surgeons, in relation to litigation. They often described how people from

these professions appeared less worried when involved in a litigation claim and did not seem to take it as personally as physiotherapists. Most physiotherapists perceived these differences were because other professions had more awareness of litigation due to having clinical negligence training within their undergraduate training. As such, other HCP felt more prepared if they were cited in a clinical negligence claim.

*"When I spoke to the orthopaedic surgeon, he wasn't worried at all. Part of that is because they do have that training and they do understand litigation. They see it as not a personal thing. They see it as just part of their job, this is what happens because of where we are, what we're doing."* (P2, physiotherapist with experience)

Physiotherapists also described how other HCP seemed more aware of the legal processes and of the support they can receive from their employer or professional organisations and insurers, such as the General Medical Council (GMC) and Medical Defence Union (MDU).

*"With GPs, it's immediately [the support], "Don't worry, because everything is fine, we are going to sort all this out, and this is how we are going to do it."* (P3, physiotherapist at risk)

## Support and training: Professional body support

Many physiotherapists referred to feeling there was a of lack of support from the UK professional body for physiotherapists, the Chartered Society of Physiotherapy (CSP). For most physiotherapists involved in litigation in this study, their first point of contact for support was the professional body. However, it appeared that most were unaware that whilst the professional body provide support for physiotherapists who are self-employed, it is the physiotherapist's employer who supports an employed physiotherapist. Due to this lack of awareness of the different roles in providing support to the employed and self-employed physiotherapist, physiotherapists often felt dissatisfied by the support they received from the professional body.

*"I have known colleagues who have gone to the Chartered Society [CSP, professional body], asking for support and help about different aspects [of litigation], and they have just not wanted to know."* (P3, physiotherapist at risk)

On occasion, due to a lack of awareness of the professional body's role in litigation, some employed physiotherapists appointed a solicitor at their own cost, to engage with the professional body to try to get support.

*"So, this guy was writing official solicitor letters to the CSP, and I was getting these bills for thousands of pounds for an hour's work."* (P20, physiotherapist with experience)

However, feedback from self-employed physiotherapists who were supported by the professional body, had been found to be positive.

*"My understanding from the feedback [from self-employed physiotherapists] is that the support they receive is great . . . the service is there to support an [self-employed] individual who is normally, very shocked, really concerned and, often really panicking about what to do or what not to do."* (P39, stakeholder professional body)

*"I contacted the CSP and said, "What do I do?" And they said, "Well, we'll put you on to the legal team" . . . the solicitor that I dealt with, she was really good."* (P35, physiotherapist at risk)

## Support and training: Improving support

Going forward, physiotherapists discussed how they think improvements can be made to the support they received. Some mentioned a more individualised approach in their workplace, ensuring that physiotherapists feel they work in an environment where they feel supported and able to talk about their worries about litigation.

> *"I mean, number one, you obviously, you need people to feel that they're in a no-blame culture, don't you? You need to feel that people are, feel safe within their employment."* (P31, physiotherapist at risk)

Some talked about using training to make litigation processes and support more transparent and known.

> *"I think that package of support should then lead to you knowing who to go and speak to. I think you need to have organisational transparency."* (P6, physiotherapist with experience)

Others discussed how more support from their professional body could have been helpful for them.

> *"I guess I would have liked my professional body to be more supportive. I think that would have been really helpful. I guess a more formal process of support."* (P2, physiotherapist with experience)

Many also talked about the need for emotional support such as debriefing, networking and buddy systems.

> *"I think a network, a confidence that you can just talk through—that [someone's] got your back, a shoulder to cry on, somebody that you can really trust, and you can have a discussion with about it, I think that's really key."* (P8, physiotherapist with experience)

There were similar discussions around implementing support helplines.

> *"I think you should have a designated person within the CSP that has some counselling background; even has some legal understanding—maybe a helpline available."* (P35, physiotherapist at risk)

## Support and training: Litigation training

Most physiotherapists believed it would be beneficial for physiotherapists to be given some basic litigation training and this could be introduced at undergraduate/pre-registration level.

> *We need training on what we can and cannot say and how we handle ourselves in these situations."* (P29, physiotherapist at risk)

> *"I think we need to link in with students and with institutes of higher education to prepare physios for the climate."* (P1, physiotherapist with experience)

However, some disagreed, saying that this may scare the physiotherapists and they may change career.

*"You're going to frighten people and I know that you've got to be aware of these things but are we then creating more fear in the junior staff who are already quite fearful."* (P9, physiotherapist at risk)

Nonetheless, most agreed that training would be beneficial for graduates to be prepared.

*"I think it probably would be a scary thing at undergraduate level. I think it would probably be a lot scarier if you're going into it fresh when there's a case involving you. I know I would much rather be taught how to document things properly and have that awareness at an undergraduate level in that safe environment, rather than when the horse has already bolted, and you're being cited in a claim against you. I think that's going to be a lot scarier."* (P15, physiotherapist with experience)

It was suggested that further litigation training could be implemented at postgraduate level or at different stages along their professional career by their employer.

*"I think that the postgrad [litigation] training needs to be there. I think it will come in the advanced practice work that's going on. . . . I think it's at different levels, different stages along the professional journey really."* (P1, physiotherapist with experience)

Many talked about the potential role for the professional body for physiotherapy (CSP) to be involved in the training.

*"I think the CSP could have some sort of role, like, an e-learning package"* (P16, physiotherapist at risk)

## Discussion

This study explored the lived experience of clinical negligence litigation in UK physiotherapy. Four key themes were identified: 'Litigation effects', 'It's feels personal', 'Learning from litigation' and 'Support and training.'

This study found that litigation can have profound effects on physiotherapists' health and wellbeing. The impact seen in this study was similar to those seen in other health professions who had been involved in litigation including, stress, anxiety, high blood pressure and insomnia [19]. The term 'second victim' acknowledges the significant impact litigation can have on the HCP, both professionally and personally, including anxiety, distress, acute stress disorder, suicidal ideation, and reduced professional confidence [19, 20]. In turn, this can lead to sickness absence, burnout, and physiotherapists leaving the profession as found in this study [20]. This has serious implications for the wellbeing of the profession and the retention of the workforce. The impact of litigation on physiotherapists' clinical practice were also comparable to those seen in other health professions with defensive medicine being practiced, whereby interventions were being undertaken not wholly based on best practice, but instead to guard the clinician against future litigation claims [10].

The findings of this study show that physiotherapists often felt litigation personally and as a personal attack on their competence and ability to do their job. This finding is consistent with that found in health professions [19]. However, for some who had previous experience of being involved in a clinical negligence case, through this experience they realised that the whole patient pathway was investigated, which helped them realise it was not personal. Therefore, if physiotherapists had more knowledge of the legal process at the outset, this could help

reduce the feelings of litigation being a personal attack on them and may mitigate some of the negative effects on their health and wellbeing.

Throughout litigation it was evident that there were opportunities for learning that could be used to make positive changes going forward. The current findings show that physiotherapists felt unprepared for litigation and often did not understand the implications of litigation and where to go for support. These findings add to previous research which found that there was a lack of clear, easily accessible information describing the process and support available for UK physiotherapists in receipt of a legal claim [8]. Furthermore, findings from the current study highlight a lack of sharing of information in relation to legal claims. This was often linked with a reticence to share experiences due to the stigma associated with litigation and the feeling of a blame culture within the profession. This study found that litigation often made physiotherapists feel embarrassed and blamed in the workplace. A blame culture was also similarly described across the midwifery and medical professions [10, 13, 19]. Our findings and those of others [10, 21], suggest that reducing blame across the profession would lead to more openness and discussion around litigation in the workplace, which is needed to allow learning from litigation to occur. It is recommended that litigation cases are shared in the workplace so that lessons can be learned, and mistakes are not repeated [14]. NHS Resolution and the Getting It Right First Time (GIRFT) report have highlighted the importance that learning from litigation claims can have on improving patient safety [14].

This study found that when physiotherapists were notified that they were involved in a claim, they generally contacted their professional body for support and information on the legal process. However, whilst self-employed physiotherapists receive legal support for clinical negligence from the professional body, employed physiotherapists are supported through the litigation process by their employer via vicarious liability [8]. Our findings suggest that when employed physiotherapists contacted the professional body, this was not clearly explained to them, which resulted in some feeling unsupported by their professional body. In contrast, feedback on the support provided by the professional body to self-employed physiotherapists was positive. Therefore, whilst the professional body appear to be providing a good level of support to self-employed physiotherapists, more information needs to be provided by them to employed physiotherapists regarding where they should seek litigation support. Additionally, emotional support in the form of a buddy system, led and co-ordinated by the professional body, could be instigated. In recognition of the impact that litigation can have on the HCP involved, a National Institute for Health and Care Research funded UK website has been developed as a resource and to provide support [9]. It signposts to sources of support available, including profession specific support, however, notable by is absence is physiotherapy.

Including clinical negligence training in the undergraduate/pre-registration curriculum could help physiotherapists feel more prepared in the event of a claim. Although in our study there was some debate as to when the most appropriate time was to implement clinical negligence training, the consensus was that this should be included in the undergraduate physiotherapy curriculum and this learning should be built upon at throughout the physiotherapists career. This is supported by work recently undertaken by the Academy of Medical Royal Colleges [22] who have developed a National Patient Safety syllabus to improve patient safety in the NHS that could be incorporated into undergraduate and postgraduate healthcare education and continuing professional development. By improving clinical negligence training and support for physiotherapists, this may help reduce the worry and uncertainty for those physiotherapists who do become involved in a claim, as they should have the knowledge of where to go for support and what is involved in a claims process. This knowledge should also ensure physiotherapists do not feel litigation is a personal attack, as they would have better knowledge of the claims process. Furthermore, improving support in the workplace and sharing

experiences could help physiotherapists talk more openly about litigation, reduce the stigma, and to learn from litigation. This will not only benefit physiotherapists, ultimately it will benefit patients by improving patient safety.

## Strengths and limitations

This qualitative study recruited a large number of participants, including physiotherapists with CES litigation experience, those at risk of litigation and several stakeholders. This provided a rich and deep understanding of the phenomenon of interest. Including participants from a range of backgrounds, allowed us to generate data from different viewpoints to create a holistic understanding of the litigation experience. However, whilst the study included physiotherapists and stakeholders who felt able to discuss their experiences around litigation, there may have been others who did not feel comfortable to do this, including those who may have left the profession as a result of their experience. Therefore, this research may not have captured the whole range of physiotherapists' experiences of litigation which is a limitation of the study.

## Conclusion

This is the first study to investigate the lived experiences of litigation in UK physiotherapists. This study found that litigation impacted physiotherapists' physical and mental wellbeing and may lead them to practice more defensively or leave the profession. Physiotherapists felt litigation was a personal attack on them and their ability to do their job. Perceptions of a 'blame culture' and perceived stigma associated with the claim, led to a lack of sharing, and learning in relation to litigation. Physiotherapists were unsure who they should contact when they found out they were cited in a claim or the support available to them. The need for emotional support for those going through a legal claim was underlined. The need for training was highlighted to understand the process of litigation and range of potential outcomes, which should be introduced during undergraduate training and built on during the physiotherapists career.

## Recommendations

To help support UK physiotherapists involved in litigation, it is recommended that:

- There should be a single repository of information describing who physiotherapists should contact if they become involved in litigation. Better signposting to profession specific support is needed.

- Emotional support in the form of a helpline and a buddy system should be instigated, which could be led and co-ordinated by the physiotherapy professional body.

- Training on clinical negligence should be introduced at undergraduate/pre-registration level for physiotherapists. Litigation training should then be implemented in more detail throughout a physiotherapists career.

- Feedback from litigation cases should be shared both locally and nationally for learning from litigation to occur and to reduce the blame culture and stigma associated with litigation.

## Supporting information

**S1 Checklist. COREQ checklist.**
(PDF)

**S1 File. Interview topic guide.**
(DOCX)

## Acknowledgments

The authors would like to thank the PPI group for their contribution to developing the interview topic guide.

## Author Contributions

**Conceptualization:** Gillian Yeowell, Susan Greenhalgh, Emma Willis, James Selfe.

**Data curation:** Gillian Yeowell.

**Formal analysis:** Gillian Yeowell, Rachel Leech, Susan Greenhalgh, Emma Willis, James Selfe.

**Funding acquisition:** Gillian Yeowell, Susan Greenhalgh, Emma Willis, James Selfe.

**Investigation:** Gillian Yeowell, Rachel Leech, Susan Greenhalgh, James Selfe.

**Methodology:** Gillian Yeowell, Rachel Leech, Susan Greenhalgh, Emma Willis, James Selfe.

**Project administration:** Gillian Yeowell, Susan Greenhalgh.

**Supervision:** Gillian Yeowell, James Selfe.

**Validation:** Gillian Yeowell, Rachel Leech, Susan Greenhalgh, Emma Willis, James Selfe.

**Writing – original draft:** Gillian Yeowell, Rachel Leech, Susan Greenhalgh, Emma Willis, James Selfe.

**Writing – review & editing:** Gillian Yeowell, Rachel Leech, Susan Greenhalgh, Emma Willis, James Selfe.

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
