## [Decision Letter · Decision Letter 0]

18 Jul 2023

PONE-D-23-17013The lived experiences of UK physiotherapists involved in Cauda Equina Syndrome litigation. A qualitative studyPLOS ONE

Dear Dr. Yeowell,

Thank you for submitting your manuscript to PLOS ONE. After careful consideration, we feel that it has merit but does not fully meet PLOS ONE’s publication criteria as it currently stands. Therefore, we invite you to submit a revised version of the manuscript that addresses the points raised during the review process.

We look forward to receiving your revised manuscript.

Kind regards,

Shabnam ShahAli, Ph.D.

Academic Editor

PLOS ONE

Reviewers' comments:

Reviewer's Responses to Questions

**Comments to the Author**

1. Is the manuscript technically sound, and do the data support the conclusions?

Reviewer #1: Yes

Reviewer #2: Yes

2. Has the statistical analysis been performed appropriately and rigorously? 

Reviewer #1: Yes

Reviewer #2: Yes

3. Have the authors made all data underlying the findings in their manuscript fully available?

Reviewer #1: Yes

Reviewer #2: Yes

4. Is the manuscript presented in an intelligible fashion and written in standard English?

Reviewer #1: Yes

Reviewer #2: Yes

5. Review Comments to the Author

Reviewer #1: Dear Authors,

This study addresses an important topic. I enjoyed reading this manuscript. This is an interesting and novel study about the litigation effect on several aspects of physiotherapist’s life and work. The authors described a robust, well-designed study and have reported some interesting results. That said, the manuscript in its present form has some shortcomings regarding the method that I have outlined below. I have also provided one specific comment for consideration.

Specific comment:

Abstract- method: Please add the COREQ as a criterion for qualitative research.

Methods

How did the themes and sub-themes refine by the research team? Please more explain about the process of identifying, selecting, and refining 4 main themes and several sub-themes.

Why didn’t you consider the litigation effects on married life, sexual life, and specially for women, the mother-child relationship? It seems they could be important and should be considered as sub-themes for litigation effect.

Reviewer #2: Dear Authors

The manuscript was well-written. Thanks for submitting your manuscript. I have checked the qualitative checklist for your manuscript. All the requirements were provided. Please make sure to avoid dual publication.

6. PLOS authors have the option to publish the peer review history of their article (what does this mean?). If published, this will include your full peer review and any attached files.

Reviewer #1: **Yes: **Mehrnaz Kajbafvala

Reviewer #2: No

---

## [Author Response · Author response to Decision Letter 0]

7 Aug 2023

Editor’s comments

Comment E1. 

Response E1

The manuscript has been checked to ensure it meets PLOS ONE's style requirements, including those for file naming.

Comment E2. 

Response E2

Data derived codes were used in the analysis of the paper. A minimal data set has been provided to facilitate reproducibility and reuse. Author generated codes were not used as part of this research.

Comment E3. 

Response E3.1

The following text has been added to the manuscript (p6) and included in the online submission:

Informed consent was obtained verbally from all participants prior to participation. Participant consent was digitally audio-recorded by the interviewers.

Response E3.2

N/A

Comment E4

Upon re-submitting your revised manuscript, please upload your study’s minimal underlying data set as either Supporting Information files or to a stable, public repository and include the relevant URLs, DOIs, or accession numbers within your revised cover letter. 

Response E4

We have uploaded the minimal data set underlying the results described in our manuscript to the university repository. This can be accessed via DOI 10.23634/MMU.00632413 (which will be made live on acceptance of the paper)

Comment E5. 

Response E5

The reference list has been reviewed to ensure it is complete and correct. 

Comment E6

6. While revising your submission, please upload your figure files to the Preflight Analysis and Conversion Engine (PACE) digital diagnostic tool, https://pacev2.apexcovantage.com/. 

Response E6

Figure files (Fig 1) have been uploaded to PACE and have been included.

Reviewers’ comments

Reviewers' comments:

Reviewer's Responses to Questions

Comments to the Author

1. Is the manuscript technically sound, and do the data support the conclusions?

Reviewer #1: Yes

Reviewer #2: Yes

Response – Thank you 

2. Has the statistical analysis been performed appropriately and rigorously? 

Reviewer #1: Yes

Reviewer #2: Yes

Response – Thank you 

3. Have the authors made all data underlying the findings in their manuscript fully available?

Reviewer #1: Yes

Reviewer #2: Yes

Response – Thank you 

4. Is the manuscript presented in an intelligible fashion and written in standard English?

Reviewer #1: Yes

Reviewer #2: Yes

Response – Thank you 

Reviewer 1 

Comment R1_5. 

5. Review Comments to the Author

Reviewer #1: Dear Authors,

This study addresses an important topic. I enjoyed reading this manuscript. This is an interesting and novel study about the litigation effect on several aspects of physiotherapist’s life and work. The authors described a robust, well-designed study and have reported some interesting results. That said, the manuscript in its present form has some shortcomings regarding the method that I have outlined below. I have also provided one specific comment for consideration.

Response – Thank you 

Comment R1_5.1

Specific comment:

Abstract- method: Please add the COREQ as a criterion for qualitative research.

Response R1_5.1

This has been added (p2)

Comment R1_5.2

Methods

How did the themes and sub-themes refine by the research team? Please more explain about the process of identifying, selecting, and refining 4 main themes and several sub-themes.

Response R1_5.2

Additional text has been added to the manuscript (p7) as follows to provide further detail re the process:

This involved the team (XX, XX, XX) independently listening to the audio-recordings and reading the transcripts. Open coding was used to code the data. This involved reading each transcript line by line to identify salient text related to the research question. Data derived codes were used to summarise the data. Codes were recorded using Nvivo software (version 20.6.1). Patterns across the dataset were then iteratively explored, and theoretically cognate codes were grouped to create sub-themes. Conceptually similar sub-themes were grouped together into emergent themes independently by the research team. The themes were then discussed, critically reviewed, and refined by the research team (XX, XX, XX, XX, XX) to create the final themes. There was concordance in the themes identified by the team and any refinement of themes related to semantics. Preliminary analysis was undertaken after each interview, which iteratively fed into subsequent data generation. 

Comment R1_5.3

Why didn’t you consider the litigation effects on married life, sexual life, and specially for women, the mother-child relationship? It seems they could be important and should be considered as sub-themes for litigation effect.

Response R1_5.3

Thank you. Gadamerian hermeneutic phenomenology was used to explore the participants’ lived experience. Using this approach, broad open questions were asked, and specific a priori questions, which may have biased the data generated, were avoided. The literature and PPI group informed the topics and the broad open questions used in the interviews. This included a broad open question about the personal impact on the participant’s wellbeing, with follow-up questions being asked in response to what the participant said, to explore their experience in greater depth. As participants in our study did not discuss or raise their married life, sexual life, and the mother-child relationship as being impacted by litigation, we could not include this as a sub-theme in this study.

Reviewer 2

Reviewer #2: 

Comment R2_6

Dear Authors 

The manuscript was well-written. Thanks for submitting your manuscript. I have checked the qualitative checklist for your manuscript. All the requirements were provided. Please make sure to avoid dual publication.

Response R2_6

Thank you

---

## [Decision Letter · Decision Letter 1]

18 Aug 2023

The lived experiences of UK physiotherapists involved in Cauda Equina Syndrome litigation. A qualitative study

PONE-D-23-17013R1

Dear Dr. Yeowell,

We’re pleased to inform you that your manuscript has been judged scientifically suitable for publication and will be formally accepted for publication once it meets all outstanding technical requirements.

Kind regards,

Shabnam ShahAli, Ph.D.

Academic Editor

PLOS ONE

1. If the authors have adequately addressed your comments raised in a previous round of review and you feel that this manuscript is now acceptable for publication, you may indicate that here to bypass the “Comments to the Author” section, enter your conflict of interest statement in the “Confidential to Editor” section, and submit your "Accept" recommendation.

Reviewer #1: All comments have been addressed

Reviewer #2: All comments have been addressed

2. Is the manuscript technically sound, and do the data support the conclusions?

Reviewer #1: Yes

Reviewer #2: Yes

3. Has the statistical analysis been performed appropriately and rigorously? 

Reviewer #1: N/A

Reviewer #2: N/A

4. Have the authors made all data underlying the findings in their manuscript fully available?

Reviewer #1: Yes

Reviewer #2: Yes

5. Is the manuscript presented in an intelligible fashion and written in standard English?

Reviewer #1: Yes

Reviewer #2: Yes

6. Review Comments to the Author

Reviewer #1: Dear Authors

Thank you for your detail and complete responses. All reviewer comments has been answered correctly.

Reviewer #2: (No Response)

7. PLOS authors have the option to publish the peer review history of their article (what does this mean?). If published, this will include your full peer review and any attached files.

Reviewer #1: **Yes: **Mehrnaz Kajbafvala

Reviewer #2: **Yes: **Zinat Ashnagar

---

## [Editor Report · Acceptance letter]

6 Sep 2023

PONE-D-23-17013R1 

The lived experiences of UK physiotherapists involved in Cauda Equina Syndrome litigation. A qualitative study 

Dear Dr. Yeowell:

I'm pleased to inform you that your manuscript has been deemed suitable for publication in PLOS ONE. Congratulations! Your manuscript is now with our production department. 

Kind regards, 

on behalf of

Dr. Shabnam ShahAli 

Academic Editor

PLOS ONE